# Joint Verification and Refinement of Language Models for Safety-Constrained Planning

## Abstract

Large language models possess impressive capabilities in generating programs (e.g., Python) from natural language descriptions to execute robotic tasks. However, these generated programs often contain errors that violate externally given task specifications. Without an effective method to verify their correctness, the reliable deployment of language models in real-world systems is practically infeasible. We develop a method that converts generated robot programs into an automaton-based representation and verifies them against task-relevant safety specifications. We establish a theorem that any arbitrary combination of the verified programs will also satisfy the safety specifications. Hence, the method eliminates the need to verify complex programs composed of multiple simpler ones, reducing computation complexity. We then introduce an automated fine-tuning procedure that leverages verification outcomes for supervision. By applying the theorem, this procedure only requires training the model to generate safe sub-components, thereby improving training efficiency. Empirical results on robot applications show a 30 percent increase in the probability of generating specification-compliant programs, with training time reduced by half compared to fine-tuning on generating full programs. Code available: https://tinyurl.com/safe-codegen.

## 1 Introduction

As large language models (LLMs) have demonstrated significant potential in generating programs for solving robot tasks Wang et al. (2023); et al. (2022; 2021); Hu et al. (2024b), the generated programs often fail to meet the externally provided task specifications, which may lead to severe consequences in safety-critical contexts. Existing approaches Hu et al. (2024b); Liu et al. (2023b); Du et al. (2024) verify the programs by empirically collecting and checking execution traces. Such empirical verification may miss corner cases that violate the specifications. Therefore, guaranteeing that the generated programs satisfy task specifications remains challenging.

Formal verification provides guarantees of compliance with specifications Baier & Katoen (2008); Mac-Conville & Monahan (2024); Shu et al. (2019); Sun et al. (2024), but applying it to LLM-generated programs presents unique challenges. Unlike programs written with well-defined patterns, LLM-generated programs exhibit flexible structures and inconsistent variable naming. These factors increase the difficulty of *program abstraction*—a key component of verification that expresses a program in a finite-state machine. Moreover, the size of such programs increases the state space, making verification computationally expensive.

We develop a program verification method that **(1)** verifies the high-level behaviors of LLM-generated programs against externally provided specifications, and **(2)** alleviates the computational complexity of verifying complex, long-horizon programs. Given a set of logical specifications (e.g., safety constraints) and an LLM-generated program (e.g., in Python), the method converts the program into an automaton that is amenable to formal verification tools, such as model checking. It *enables verification of programs with flexible structures and varied lengths.* To address scalability, we establish a **compositional verification theorem**: if individual program components each satisfy the safety specification, then any arbitrary composition of these components also satisfies it. This theorem (illustrated in fig. 1) modularizes the verification process, *reduces the need for exhaustive, system-wide verification* while maintaining safety guarantees.

In addition, we introduce an automated fine-tuning procedure that leverages verification outcomes to improve the LLM's ability to generate specification-compliant programs without requiring human annotations. The procedure treats programs that pass formal verification as positive training examples and updates the model parameters in a supervised manner. Importantly, the compositional verification theorem enhances fine-tuning efficiency: Instead of training the LLM to generate entire programs, we only need to fine-tune it to generate sub-components that meet safety specifications. Through this process, we achieve a **30 percent** probability of generating specification-satisfying programs with only 20 minutes of training—**halving the time** required compared to training for full program generation.

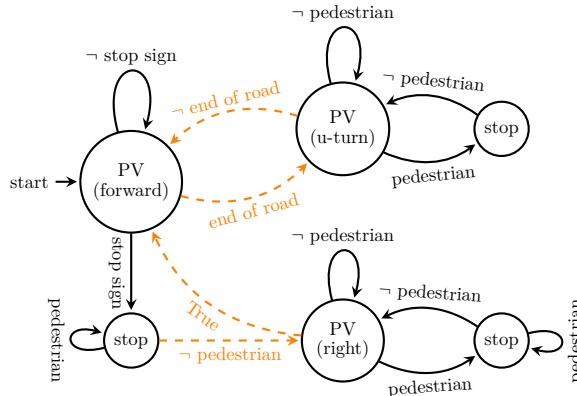

Figure 1: A composed program of three subprograms. The states of each subprogram are connected in black transitions. The orange dashed transitions connect the subprograms. We establish a theorem that any combination of individually verified subprograms also satisfies the safety specifications.

In summary, the contributions of this work are twofold:

- **Verification:** Develop a method to verify the safety of the LLM-generated programs, along with a theorem that ensures the safety of complex, long-horizon programs.

- **Learning:** Propose a fine-tuning procedure that improves specification compliance without human labels, achieving better performance with reduced training time by leveraging the compositional verification theorem.

## 2 Related Work

Many existing works develop methods to generate executable programs (e.g., C++ or Python) via language models Svyatkovskiy et al. (2020); Fried et al. (2022); Rozière & et al. (2023); et al. (2023); Nijkamp et al. (2023); Wang et al. (2023); et al. (2022; 2021); Ahmad et al. (2021); Singh et al. (2022). However, these works lack verification of their generated programs and directly execute them, which is risky in safety-critical applications. The works Hu et al. (2024b;a); et al. (2022; 2021); Liu et al. (2023b); Hendrycks et al. (2021); Austin et al. (2021); Du et al. (2024); Nguyen & Nadi (2022); Ni et al. (2023) empirically verify generated programs against externally provided specifications. However, such empirical tests may miss edge cases, which still pose a safety risk. In contrast, our proposed method provides formal guarantees, ensuring the program satisfies given specifications in all possible scenarios, including all the edge cases.

Traditional program verification methods can verify program behaviors via model checking Nelson (1980); Hoare (1969); Pnueli (1977); Clarke et al. (2018); Vardi & Wolper (1986); Kurshan (2000); Farias et al. (2024) or transform programs into formal languages to constrain the values of variables or check runtime errors, e.g., dividing by 0 MacConville & Monahan (2024); Shu et al. (2019); Sun et al. (2024). Meanwhile, recent work has made progress in verifying high-level plans expressed in natural language Liu et al. (2023a); Yang et al. (2024); Yang & et al. (2024). However, these methods are incapable of verifying LLM-generated programs with unconventional constructs, inconsistent naming, or non-modular logic, and are computationally expensive when verifying long-horizon programs.

## 3 Problem Formulation

Consider a system $\mathcal{S} = (S, E, AP_S, AP_E, \Phi)$ provided by a system designer, where

- $S$ is a set of *subscribing functions* (API calls) receiving and extracting environment or system information. Each subscribing function $f_s \in S$ takes inputs from text space $\mathcal{T}$ (a set of all possible texts) and returns a Boolean value, i.e., $f_s : \mathcal{T} \to \{0, 1\}$.

- $E$ is a set of *execution functions* that publish actions for the system to execute.

- $AP_S$ is a set of atomic propositions corresponding to $S$. Each function $f_s \in S$ corresponds to a proposition in $AP_S$.

- $AP_E$ is a set of atomic propositions corresponding to functions in $E$.

- $F_C : S \cup E \to AP_S \cup AP_E$ maps a function (with its input and output) to a corresponding atomic proposition.

- $\Phi$ is a set of *safety specifications* over $AP_S$ and $AP_E$.

A safety specification is a temporal logic formula Rescher & Urquhart (2012) asserts that "bad things" never happen during a system's execution.

Let $P$ be a language model-generated program formally defined as:

**Definition 1.** A PROGRAM $P$ is a computer program describing a set of function sequences. Each sequence $f_1 f_2 ...$ consists of functions $f_i \in S \cup E$ for $i = 1, 2, ....$.

We show some generated programs in Section 5.1. Then, we can formulate our problem.

**Problem 1:** Given a system $\mathcal{S} = (S, E, AP_S, AP_E, F_C, \Phi)$ and a program $P$, **formally verify** whether $P$ satisfies the safety specifications $\Phi$.

Once we can verify a single program against safety specifications, we want to prove that any combination of verified programs will still satisfy the safety specifications. Let $\{P_i\}_{i=1}^m$ be a set of $m$ programs. We can combine these programs to solve complex tasks.

**Definition 2.** A COMPOSED PROGRAM $\mathcal{C}_p$ OF $\{P_i\}_{i=1}^m$ is a sequence of programs $P_1^C P_2^C P_3^C ...,$ $\forall_{j \in \mathbb{N}} \; P_j^C \in \{P_i\}_{i=1}^m$.

We show an example of a composed program $\mathcal{C}_p$ in Fig. 1.

A composed program $\mathcal{C}_p$ describes a set of function sequences, where each sequence is a concatenation of sequences described by programs in $P_1^C P_2^C P_3^C ....$. For example, if $f_1 f_2$ and $f_a f_b f_c$ are sequences described by $P_1^C$ and $P_2^C$, respectively, then $f_1 f_a f_2 f_b ...$ is in $\mathcal{C}_p$. **Note** that a composed program does not need to complete $P_1^C$ and then transit to the beginning of $P_2^C$. Instead, it can halt $P_1^C$ at any point and transit to any point at $P_2^C$.

**Problem 2:** Given a system $\mathcal{S} = (S, E, AP_S, AP_E, F_C, \Phi)$, let $\mathcal{C}_p$ be a composed program of $\{P_i\}_{i=1}^m$, prove the following statement:

*If every program in $\{P_i\}_{i=1}^m$ satisfies $\Phi$, then the composed program $\mathcal{C}_p$ of $\{P_i\}_{i=1}^m$ also satisfies $\Phi$.*

## 4 Methodology

We present a method for verifying and improving LLM-generated robot programs to satisfy externally provided safety specifications. The method consists of three key components: **(1)** converting the generated program into an automaton-based representation suitable for formal verification, **(2)** applying a compositional verification theorem to scale verification across program components, and **(3)** fine-tuning the language model using verified subprograms to improve specification compliance.

### 4.1 Program Verification Against Safety Specifications

Since a program is not formally verifiable directly, we transform it into a verifiable form and then verify the transformed program against logic-based safety specifications in $\Phi$.

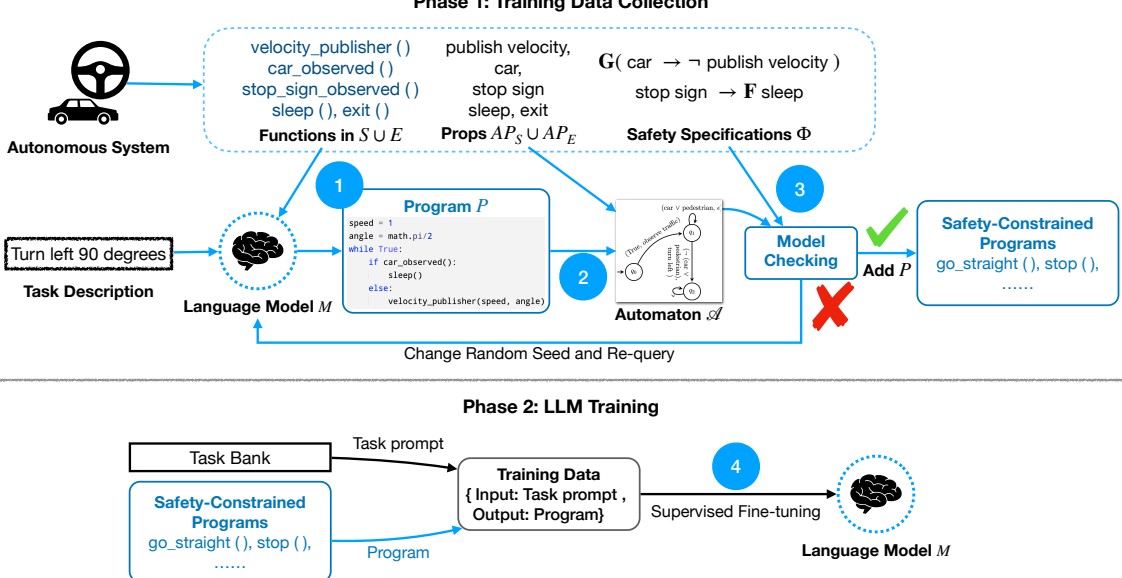

Figure 2: In this pipeline, we verify each LLM-generated program and add the specification-compliant programs to a set of *safety-constrained programs*, which will be used for fine-tuning the LLM.

First, we construct a *transition system* $TS = (Q_s, T_s, L_s)$ representing the task environment.

**Definition 3.** A **transition system** $TS = (Q_s, T_s, L_s)$ is a tuple of a set of states $Q_s$, a set of transitions $T_s = \{(q_i, q_j) \mid q_i, q_j \in Q_s\}$, i.e., $(q_i, q_j)$ means a transition from state $q_i$ to $q_j$, and a label function $L_s : Q_s \to 2^{AP}$.

$AP$ is a set of atomic propositions. Each atomic proposition has a truth value—true or false—but does not contain any logical connectives like "and," "or," "not," etc. This system builds transitions between every conjunction of the truth values of propositions in $AP_S$.

Next, we construct an automaton that represents each program.

**Definition 4.** A **finite-state automaton (FSA)** $\mathcal{A} = (Q_a, p_0, T_a, L_a)$ is a tuple consisting of a set of states $Q_a$, an initial state $p_0$, a set of transitions $T_a = \{(p_i, \sigma, p_j) \mid p_i, p_j \in Q_a, \sigma \in 2^{AP}\}$, and a label function $L_a : Q_a \to 2^{AP}$.

Consider a system $\mathcal{S}$, a program $P_i$, and a transition system $TS$. We construct an FSA $\mathcal{A}$ such that: For every function sequence $f_1 f_2, ...$ described by $P_i$, there is a corresponding sequence $F_C(f_1) F_C(f_2)...$ described by $\mathcal{A}$.

To build the FSA, we follow the three steps below:

1) Parse $P_i$ into an abstract syntax tree (AST).

2) Define a keyword processor that maps an AST with predefined keywords and specified structures to an FSA, as presented in Table 1.

3) Follow Algorithm 1 to build an FSA. Note that the algorithm takes the root of AST and the keyword processor as inputs and returns an FSA $\mathcal{A} = (Q_a, p_0, T_a, L_a)$.

Once the FSA $\mathcal{A}$ is constructed, we formulate a *product automaton* by implementing the FSA in the transition system $TS$, denoted as $\mathcal{P} = \mathcal{A} \otimes TS$.

**Definition 5.** Given an FSA $\mathcal{A}$ and a transition system $TS$, a **product automaton** $\mathcal{P}$ of $\mathcal{A}$ and $TS$, denoted $\mathcal{P} = \mathcal{A} \otimes TS$, is a tuple $(Q, Q_0, T, L)$, where

---

Algorithm 1: AST to *Finite state automaton* (FSA)

---

1: **procedure Tree2FSA**(root, keywords, keyword_processor)    ▷ *keywords* is a set of predefined words, *keyword_processor* is a function
2:     $Q_a, T_a, L_a = [], [], []$
3:     create an initial state $p_0$, $Q_a$.add($p_0$), $L_a(p_0) = \emptyset$
4:     $p_{current} = p_0$                                                                          ▷ keep track of the current state
5:     **for** node in root.children **do**
6:         **if** (every node in node.children is leaf) | (node.children[0] in keywords) **then**
7:             $\tilde{Q}, \tilde{p}_0, \tilde{T}, \tilde{L} = $ keyword_processor(node)
8:         **else**
9:             $\tilde{Q}, \tilde{p}_0, \tilde{T}, \tilde{L} = $ **Tree2FSA**(node, keywords, keyword_processor)        ▷ Preorder Traversal
10:         **end if**
11:         $Q_a+ = \tilde{Q}, T_a+ = \tilde{T}, L_a+ = \tilde{L}$                                      ▷ merge the sub-automaton
12:         $T_a$.add(($p_{current}, True, \tilde{p}_0$))
13:         $p_{current} = \tilde{p}_0$
14:     **end for**
15:     **return** $Q_a, p_0, T_a, L_a$
16: **end procedure**

---

$Q = \{(p, q) \mid p \in Q_a, q \in Q_s\}$, $Q_0 = \{p_0\} \times Q_s$,    $T = \{((p, q), (p', q')) \mid p \in Q_a, q \in Q_s, (p, L_s(q), p') \in T_a, (q, q') \in T_s\}$,    and $L((p, q)) = L_a(p) \cup L_s(q)$, where $p \in Q_a, q \in Q_s$.

In the product automaton $\mathcal{P} = (Q, Q_0, T, L)$,

- a **prefix** is a finite sequence of states starting from $(p_0, q_0) \in Q_0$, e.g., $(p_0, q_0)(p_1, q_1)(p_2, q_2)...(p_k, q_k)$, $k$ is the prefix length,

- a **trace** $\phi$ is a sequence of labels $L((p_0, q_0))L((p_1, q_1))\dots$, where Traces($\mathcal{P}$) denotes the set of all traces from $\mathcal{P}$. In words, Traces($\mathcal{P}$) captures all possible behaviors of $\mathcal{A}$ in the task environment represented by $TS$.

Finally, we use a model checker Cimatti et al. (2002) to verify whether the product $\mathcal{P}$ satisfies all the specifications $\Phi$. If a program's automaton satisfies all the specifications, we add this program to a set of *safety-constrained programs*. We present an illustration of this procedure in fig. 2 Phase 1.

### 4.2   Safety of Composed Program

As verifying long-horizon programs directly can be computationally expensive, we propose a *compositional verification theorem* that allows the safety of complex programs to be inferred from their components. Specifically, if each subprogram satisfies the safety specification in isolation, then their composition also satisfies the specification under mild assumptions. An example of a composed program is in fig. 1.

To establish and prove the theorem, we first need to define the terminology *safety*. Let $\phi \in \Phi$ be a temporal logic formula. We call $\phi$ a *safety specification* if it describes a *safety property* Baier & Katoen (2008) as defined in definition 6.

**Definition 6.** A **safety property** $P_{\text{safe}}$ is a set of traces in $(2^{AP})^\omega$ ($\omega$ means infinite repetitions) such that for all traces $\psi \in (2^{AP})^\omega \backslash P_{\text{safe}}$, there is a finite-length prefix $\hat{\psi}$ such that

$$P_{\text{safe}} \cap \{\psi \in (2^{AP})^\omega \mid \hat{\psi} \text{ is a prefix of } \psi\} = \emptyset.$$

$\hat{\psi}$ is a bad prefix, and BadPref($P_{\text{safe}}$) denotes the set of all bad prefixes.

From the definition of safety property, we derive the proposition below.

| AST | FSA | Note |
|---|---|---|
| start → root, while, $f_s$, $f_e$ | start → $\emptyset$, $\omega$ | $\sigma = F_C(f_s)$, $\omega = F_C(f_e)$, $f_s \in S, f_e \in E$. "For loop" can be expressed by "while loop." |
| start → root, if, $f_s$, $f_e$ | start → $\emptyset$, $\omega$ | $\sigma, \omega = F_C(f_s), F_C(f_e)$, $f_s \in S, f_e \in E$. |
| start → root, if, else, $f_s$, $f_{e1}$, $f_{e2}$ | start → $\emptyset$, $\omega_1$, $\omega_2$ | $\sigma, \omega_1, \omega_2 = F_C(f_s)$, $F_C(f_{e_1}), F_C(f_{e_2})$. For "if-elif-else," we duplicate the "if" node and replace it with "elif." |

Table 1: A subset of rules to convert abstract syntax trees to FSA-based representations. We present the complete rules in the Appendix.

**Proposition 1.** Let $\phi$ describe safety property $P_{\text{safe}}$, an automaton $\mathcal{P}$ satisfies $\phi$ (denoted as $\mathcal{P} \models \phi$) if and only if $\text{Traces}(\mathcal{P}) \subseteq P_{\text{safe}}$.

From section 4.1, we obtain a set of safety-constrained programs where each program satisfies the safety specifications. To proceed with our theorem establishment, we introduce a terminology *joint automaton* as defined in Definition 7.

**Definition 7.** Let $\mathcal{P}_1 = (Q_1, Q_{0_1}, T_1, L_1)$ and $\mathcal{P}_2 = (Q_2, Q_{0_2}, T_2, L_2)$ be two automata over the same set of atomic propositions. Consider a new set of transitions $T^* : \{(q, q') \mid q \in Q_1, q' \in Q_{0_2}\}$ that transit from a subset of $\mathcal{P}_1$'s states to a subset of $\mathcal{P}_2$'s initial states. We define $\mathcal{P}^* = (Q_1 \cup Q_2, Q_{0_1}, T_1 \cup T_2 \cup T^*, L_1 \cup L_2)$ as a JOINT AUTOMATON of $\mathcal{P}_1$ and $\mathcal{P}_2$.

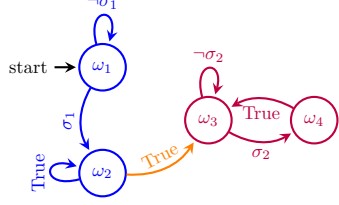

Figure 3: An example of a joint automaton $\mathcal{P}^* = (Q_1 \cup Q_2, Q_{0_1}, T_1 \cup T_2 \cup T^*, L_1 \cup L_2)$ of $\mathcal{P}_1$ and $\mathcal{P}_2$. We mark $\mathcal{P}_1$ and $\mathcal{P}_2$ in blue and purple, and mark the transition in $T^*$ in orange.

Figure 3 is a joint automaton of two automata.

Note that we can "connect" a joint automaton of $\mathcal{P}_1$ and $\mathcal{P}_2$ with $\mathcal{P}_3$ to obtain a new joint automaton of the three automata. By repeating this procedure, we can get the joint automaton of any number of automata. Such a joint automaton is the representation of the composed program:

**Remark 1.** Let $\{P_i\}_{i=1}^m$ be a set of safety-constrained programs, $\{\mathcal{P}_i\}_{i=1}^m$ be the product automata corresponding to the programs. Let $\mathcal{C}_p$ be a composed program of the programs in $\{P_i\}_{i=1}^m$, then there exist a joint automaton $\mathcal{P}^*$ captures the behaviors of $\mathcal{C}_p$.

**Theorem 1. [Compositional Verification Theorem]** Given a safety property $P_{\text{safe}}$, two automata $\mathcal{P}_1 = (Q_1, Q_{0_1}, T_1, L_1)$ and $\mathcal{P}_2 = (Q_2, Q_{0_2}, T_2, L_2)$, let $\mathcal{P}^* = (Q_1 \cup Q_2, Q_{0_1}, T_1 \cup T_2 \cup T^*, L_1 \cup L_2)$ be a joint automaton of $\mathcal{P}_1$ and $\mathcal{P}_2$, assume

1) $\mathcal{P}_1$ and $\mathcal{P}_2$ satisfy $P_{\text{safe}}$,

2) for any prefix $\hat{\psi} \notin \text{BadPref}(P_{\text{safe}})$, for any $(q, q') \in T^*$,

$$\hat{\psi}L_1(q)L_2(q') \notin \text{BadPref}(P_{\text{safe}}), \tag{1}$$

then $\mathcal{P}^*$ satisfies $P_{\text{safe}}$.

*Proof.* Assume $\mathcal{P}^*$ does not satisfy $P_{\text{safe}}$, there exists a trace $\psi$ from $\mathcal{P}^*$ such that $\psi$ has a prefix $\hat{\psi} \in \text{BadPref}(P_{\text{safe}})$.

Let $\psi = \psi_1 L_1(q) L_2(q') \psi_2$ be a trace with the bad prefix, where $\psi_i \in \text{Traces}(\mathcal{P}_i), i \in [1, 2]$ and $(q, q') \in T^*$.

Since $\mathcal{P}_1$ satisfies $P_{\text{safe}}$, $\psi_1$ does not contain any bad prefix. Then, by the assumption of eq. (1), $\psi_1 L_1(q) L_2(q')$ does not contain any bad prefix. Similarly, $\psi_2$ does not contain bad prefix because $\mathcal{P}_2$ satisfies $P_{\text{safe}}$.

Therefore, $\psi$ does not have a bad prefix, which leads to a contradiction. Hence, we have proved that $\mathcal{P}^*$ satisfies $P_{\text{safe}}$. □

**Proposition 2.** Given a safety property $P_{\text{safe}}$, let $\mathcal{P}^*$ be a joint automaton of $\{\mathcal{P}_i\}_{i=1}^m$ such that

- all $\mathcal{P}_i, i \in [1, ..., m]$ satisfy $P_{\text{safe}}$,

- for any prefix $\hat{\psi} \notin \text{BadPref}(P_{\text{safe}})$, for any $(q, q')$ such that $q \in Q_x, q' \in Q_{0_y}, x \neq y$,     eq. (1) holds,

then, $\mathcal{P}^*$ satisfies $P_{\text{safe}}$.

*Proof.* We prove proposition 2 by induction.

Base case: the joint automaton of two automata satisfies $P_{\text{safe}}$, by theorem 1.

Inductive step: assume the joint automaton $\mathcal{P}^*$ of $m$ automata $\{\mathcal{P}_i\}_{i=1}^m$ satisfies $P_{\text{safe}}$. Consider a new joint automaton $\mathcal{P}^{**}$ of $\mathcal{P}^*$ and $\mathcal{P}_{m+1}$, where $\mathcal{P}_{m+1}$ also satisfies $P_{\text{safe}}$, by theorem 1, $\mathcal{P}^{**}$ satisfies $P_{\text{safe}}$.

By the theory of induction, we have proved proposition 2. □

For any complex task that can be broken down into simpler sub-components, it is unnecessary to construct and verify an automaton for the overall program. Instead, **the safety of the complex task can be asserted if the simpler components from which it is composed are themselves safe**. This conclusion significantly reduces verification complexity.

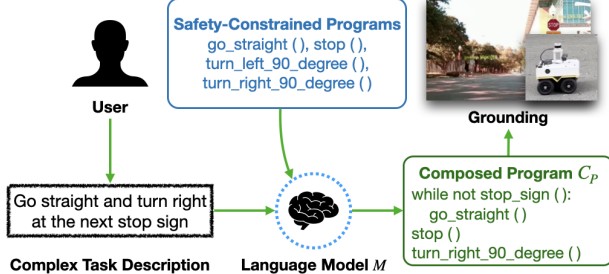

### 4.3 Verification-Guided Refinement

To improve the quality of LLM-generated programs, we introduce an automated fine-tuning procedure that uses verification outcomes as feedback. Unlike traditional supervised learning, which requires manually labeled data, our method relies entirely on formal verification results to supervise the training process. This enables the LLM to iteratively improve its ability to generate specification-compliant programs without human intervention.

Figure 4: By Theorem 1, we can safely execute the composed program to solve complex tasks without further verification.

The fine-tuning procedure works as follows:

(1) Given a set $\{t_1, t_2, ...\}$ of task descriptions, query the language model $M$ to generate executable program $\{P_1, P_2, ...\}$. We can get multiple programs with each task description by varying the random seeds.

(2) For each program $P_i$, construct an FSA $\mathcal{A}_i$ and verify it against the specifications $\Phi$.

(3) If $\mathcal{A}_i$ satisfies all the specifications, we add $P_i$ to the set of safety-constrained programs and formulate a $(t_i, P_i)$ pair that consists of the program and its corresponding task description.

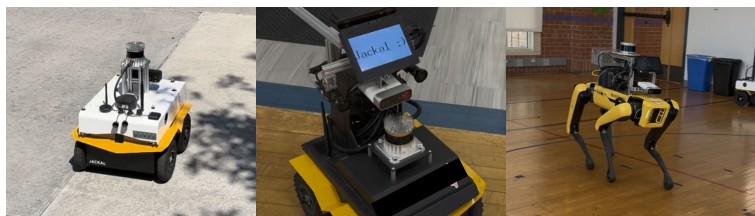

Figure 5: The three robots we used in the experiments. From left to right, we name them *Jackal outdoor robot*, *Jackal indoor robot*, and *Spot robot dog*.

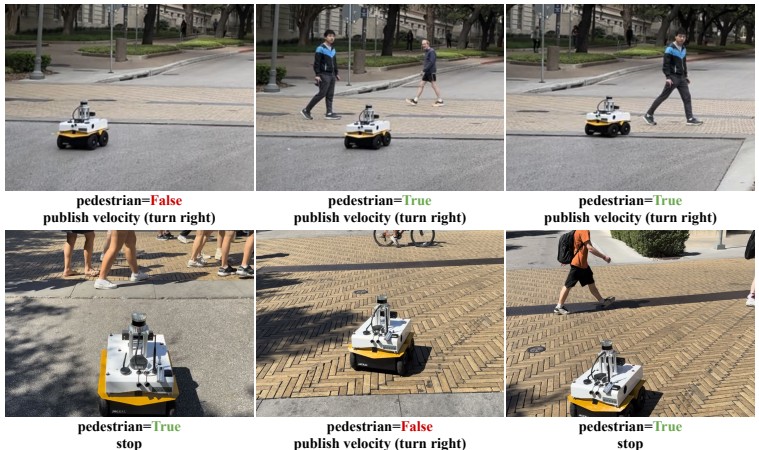

Figure 6: A failure example of executing the first program "turn_right_90_degrees_1" (top row) and a success example of executing the second program "turn_right_90_degrees_2"(bottom row). The first program publishes velocity even if a pedestrian is observed, which violates the safety specification.

(4) Repeat 2 and 3 to obtain a set of $\{(t_i, P_i)\}_{i=1}^n$ pairs, which we considered as the training dataset with $n$ samples.

(5) Use the set of pairs as supervised training data to fine-tune the language model, as presented in fig. 2 Phase 2. We consider task descriptions $t_i$ as inputs and safety-constraint programs $P_i$ as labels.

Due to the compositional verification theorem, we only need to **verify and fine-tune on subprogram generation** (each safety-constrained program is a subprogram for a composed program), **rather than training for generating complete, long-horizon programs.** This reduces the computational cost of both verification and model training.

## 5 Demonstration

We first present two robot demonstrations to iterate the steps of verifying the language model-generated programs against safety specifications in Section 5.1 and 5.2. In the experiments, we use *GPT-4o-mini* as the language model. We also demonstrate the necessity of these verification steps through two examples.

### 5.1 Outdoor Driving Task

We first present a demonstration of a *Jackal outdoor robot* (on the left of Figure 5) over a driving task. We formally define the system for this robot as follows:

$S = \{pedestrian\_observed()\}$, $E = velocity\_publisher(), stop()$,

$AP_S = \{pedestrian\}$, $AP_E = \{publish\ velocity,\ stop\}$,

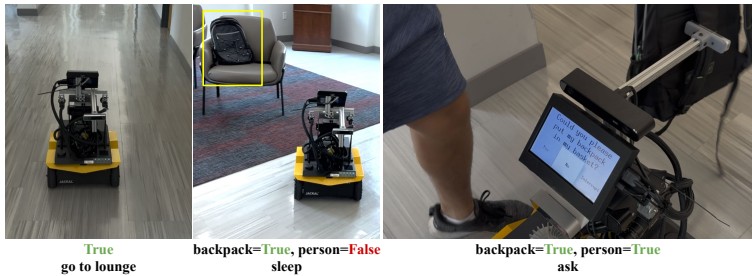

Figure 7: Execution of the program "bring_backpack_2," which passes the safety specification.

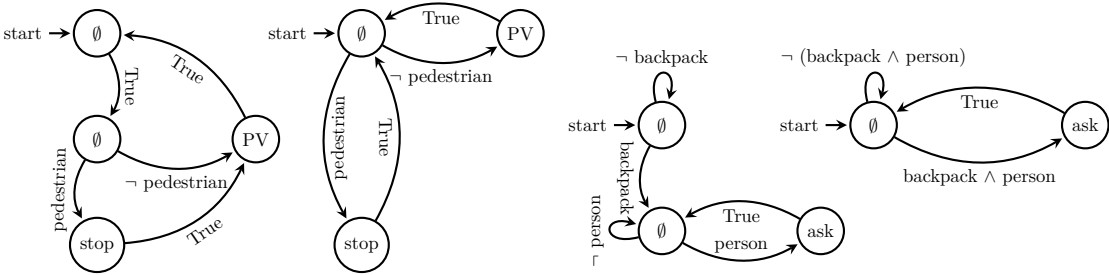

Figure 8: We show the constructed automaton-based representations of the executable programs "turn_right_90_degrees_1," "turn_right_90_degrees_2," "bring_backpack_1," and "bring_backpack_2"(from left to right). PV stands for publishing velocity.

$$F_C(pedestrian\_observed()) = pedestrian, \ F_C(stop()) = stop, \ F_C(velocity\_publisher()) = publish \ velocity,$$

and we verify the generated programs against the specification

$$\phi = \mathbf{G}( \ pedestrian \ \rightarrow \mathbf{X} \neg \ publish \ velocity \ ),$$

meaning that the system should never publish velocity when seeing a pedestrian ahead.

We send the sets of subscribing functions $S$ and execution functions $E$ (i.e., robot APIs) along with their textual descriptions to a language model. Then, query for a task "turn right at a 90-degree intersection." By varying the random seeds of the language model, we obtain the following two responses:

```
def turn_right_90_degrees_1():
    ......
    if pedestrian_observed():
        stop()
    velocity_publisher(linear, angular)
```

```
def turn_right_90_degrees_2():
    ......
    while True:
      if pedestrian_observed():
        stop()
      else:
        velocity_publisher(linear, angular)
```

Then, we follow the method in Section 4.1 to construct an automaton-based representation for each of the executable programs and present them in Figure 8. For brevity, the automata we present correspond to the blue parts in the programs. The rest are variable assignments, which are irrelevant to our specification.

Next, we verify the two automata against our safety specification $\phi$. The verification results indicate that the first program fails the specification. The counterexample shows a scenario where another pedestrian is coming after the action "stop." There is no double check on pedestrians before publishing velocity. Hence, this program fails the specification and may lead to safety risks during execution. We present an example

of such a safety violation in Figure 6. In contrast, the second program satisfies the specification and leads to a safe execution, as presented in Figure 6.

This example indicates the necessity of our proposed method. The formal verification provides mathematical guarantees for the programs. Hence, we can catch all the edge cases that may violate safety specifications without an empirical study.

### 5.2 CodeBotler

The second demonstration uses the *Jackal indoor robot* (the middle robot in Figure 5). The robot system is

$S = \{is\_in\_room(), get\_current\_location()\}, E = \{ask(), go\_to()\}$,

$AP_S = \{person, backpack\}, AP_E = \{ask, go\}$,

$F_C(is\_in\_room(\text{"person"})) = person, F_C(is\_in\_room(\text{"backpack"})) = backpack, F_C(ask(...)) = ask, F_C(go\_to(...)) = go$.

We generate programs using CodeBotler Hu et al. (2024b)—a few-shot program generator using language models—and verify the generated programs against the specification

$$\phi = \mathbf{G}(\neg(\text{ person } \wedge \text{ backpack }) \to \neg \text{ ask }).$$

We require the robot to only ask for help when the backpack and the person exist.

We query the language model to generate a program for the task "bring my backpack back from the lounge" given the APIs in $S \cup E$. We show two of the generated programs in the Appendix.

We construct automaton-based representations for the two programs and present them in Figure 8. Then, we formally verify the two automata against the specification $\phi$. The first automaton violates the specification with a counterexample "¬ backpack ∧ ask." This counterexample captures an edge case: A person takes the backpack and responds "no," the robot will ask the next person to put the backpack without checking if the backpack still exists. We argue that this edge case is hard to catch by empirical experiments, but it will lead to a specification violation. We use this example to highlight the necessity of our proposed method. In contrast, the second automaton satisfies the specification. We successfully execute the program and show the execution in Figure 7.

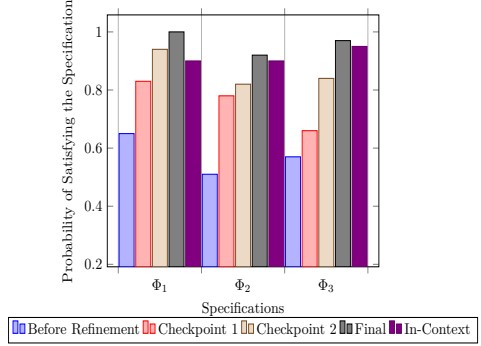

Figure 9: Probability of each specification being satisfied before and after fine-tuning the language model. Checkpoints 1, 2, and Final refer to the language model after 130, 230, and 350 epochs of fine-tuning. "In-context" refers to providing one in-context example in the queries to the language model, which serves as a baseline.

## 6 Quantitative Analysis

We have demonstrated the proposed method in the previous section and indicated its necessity. In this section, we conduct quantitative studies to show the probability of the language model generating safety-constrained programs. Then, we fine-tune the language model and show how much the fine-tuning procedure can improve such probability.

### 6.1 Automated Refinement

We first follow the steps in Section 4.3 to automatically collect fine-tuning data and use it to fine-tune the parameters of the language model. Recall that we consider the programs that pass all the specifications as the ground truth during fine-tuning. We use the system described in Section 5.1 and the following specifications to fine-tune the language model:

$\phi_1 = \mathbf{G}(\text{ pedestrian } \to \mathbf{X} \neg \text{ publish velocity })$,

$\phi_2 = \mathbf{G}(\neg(\text{ pedestrian } \vee \neg \text{ stop sign }) \to \mathbf{X} \neg \text{ stop }),$

$\phi_3 = \mathbf{G}(\text{ car } \to \mathbf{X} \neg \text{ publish velocity }).$

We employ the default supervised fine-tuning algorithm with negative log-likelihood loss and early stopping (at convergence) Dodge et al. (2020), as proposed by OpenAI Liu et al. (2023c). We collect 100 training samples and set the maximum number of epochs to 400. Each training sample is a (prompt, program) pair, where the prompt is a random driving task, e.g., go straight 10 meters, make a 60-degree left turn, etc.

Then, we select three checkpoints, test them over a separate set of driving tasks, and show the probability of each checkpoint generating safety-constrained programs in Figure 9. We observe a consistent improvement in the probability of satisfying each specification during fine-tuning. The final fine-tuned model **increases such probability by over 50 percent** compared with the initial model. On the other hand, our fine-tuned model also outperforms in-context learning, in which we provide between 1 and 5 manually generated in-context examples in the input prompt.

In conclusion, even in the absence of task or system knowledge, i.e., unable to provide in-context examples, our fine-tuning procedure can improve the probability of the language model generating safety-constrained programs to nearly 100 percent. In addition, this fine-tuning procedure only consumes 100 samples and **less than 5 minutes of training** on a single Nvidia A100 GPU.

By leveraging the compositional verification theorem, which ensures that any combination of verified sub-programs preserves safety, we shift the focus to fine-tuning on smaller, verifiably safe components. This modular approach reduces training time by **50 percent** while maintaining the likelihood of generating safety-compliant programs.

## 6.2 Out-of-Domain Validation

Next, we validate our fine-tuned language model over some out-of-domain autonomous systems and tasks. We validate the model via the Jackal indoor robot and Spot robot dog (see Figure 5). We have defined the system for the Jackal indoor robot in Section 5.2, and the specification is

$\phi_4 = \mathbf{G}(\neg(\text{ person } \wedge \text{ backpack }) \to \neg \text{ ask }).$

The system for the robot dog is

$S = \{person\_observed(), \ target\_observed()\},$

$E = \{navigate(), \ stop(), \ signal()\},$

$AP_S = \{person, \ target\},$

$AP_E = \{navigate, \ stop, \ signal\},$

$F_C(person\_observed()) = person, \ F_C(stop()) = stop,$

$F_C(target\_observed()) = target, \ F_C(navigate()) = navigate, \ F_C(signal()) = signal.$

The specifications for the robot dog are:

$\phi_5 = \mathbf{G}(\text{ person } \to \mathbf{X} \neg \text{ navigate }),$

$\phi_6 = \mathbf{G}(\neg \text{ person } \wedge \text{ target } \to \mathbf{X} \neg \text{ navigate }),$

$\phi_7 = \mathbf{G}(\neg \text{ target } \to \mathbf{X} \neg \text{ signal }).$

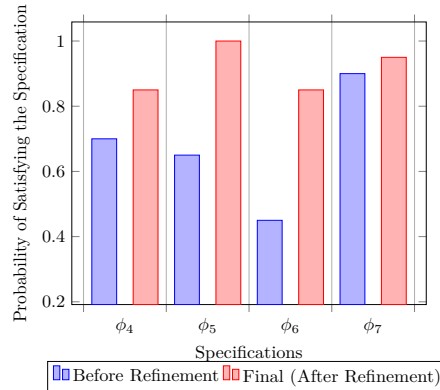

Figure 10: Out-of-domain test: We fine-tune the language model over the ground robot to meet $\phi_1, ..., \phi_4$ and then test it over a different robot (robot dog) against specifications $\phi_5, ..., \phi_7$. Over the new robot, there is an improvement in the probability of each specification being satisfied after the fine-tuning process.

We query the language model to generate 20 programs per task. The task for the indoor robot is "bringing my backpack back from the lounge," and the task for the robot dog is "finding the target and sending a signal." We compare the probability of the generated programs satisfying the specifications before and after

fine-tuning. The results in Figure 10 indicate that our fine-tuned model **improves such probability by an average of 30 percent over the out-of-domain tasks.** Hence, our fine-tuning procedure is not restricted to the system it is fine-tuned for, and it also increases the chance of satisfying safety specifications for tasks in any robot system.

### 6.3 Additional Baseline Comparison

We evaluate our fine-tuned GPT-4o-mini model against several strong pretrained baselines, including GPT-4.1, GPT-4o, GPT-4o-mini (pre-trained), and DS-R1-Qwen-7B. Despite being fine-tuned on only 100 verified programs and trained for less than one hour, our model outperforms all baselines — achieving the highest specification-satisfaction rate — while also surpassing large pre-trained models with 20 times the parameter size.

In addition to the satisfaction rate, our fine-tuned model exhibits significantly lower latency. The average response time for generating a single program is approximately half that of larger models such as GPT-4o and GPT-4.1, demonstrating the practicality of our approach for real-time applications. We present detailed results in the Appendix.

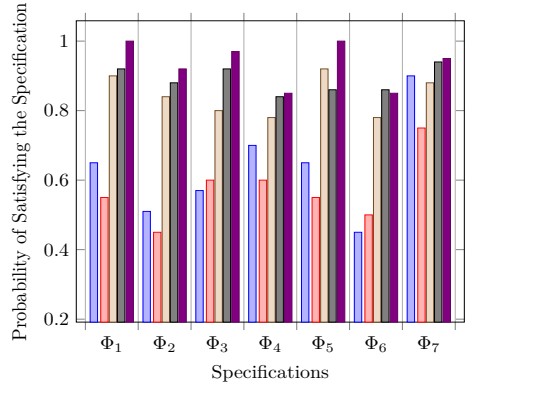

Figure 11: We compare our fine-tuned GPT-4o-mini with other pre-trained models. With only 100 training samples and less than an hour of training, our fine-tuned model outperforms baseline models with 20x parameter size.

## 7 Conclusion

This work addresses a critical limitation in deploying large language models for robotic programming by introducing a verification framework that ensures compliance with safety specifications. By representing generated programs as automata and verifying them at the sub-component level, the method guarantees the correctness of both individual and composed programs. A central contribution is the compositional verification theorem, which guarantees that any combination of individually verified sub-programs will also satisfy the overall safety specifications. This theorem removes the need to verify complex programs in their entirety, significantly reducing computational overhead.

We then propose an automated fine-tuning procedure that uses verification outcomes as supervision. Due to the composition theorem, the model only needs to learn to generate safe sub-components, enabling more efficient and modular training. Empirical results validate the effectiveness of this approach, showing a 30% increase in the generation of specification-compliant programs and a 50% reduction in training time. Together, these contributions pave the way for safer and more scalable deployment of language models in real-world robotic systems.

As a future direction, we can 1) incorporate multimodal inputs, such as visual or sensory data, into the planning process to create richer, more context-aware plans, and 2) develop systems that allow for humans-AI collaboration in program generation, where human feedback can dynamically influence the planning process to ensure compliance with nuanced or unstructured task specifications.

### Broader Impact Statement

This work enables safer and more efficient use of language models in robot program generation by developing a method to verify generated programs against task-specific safety specifications. This work guarantees that any combination of verified sub-programs also satisfies the specifications, eliminating the need to verify complex compositions and improving computational efficiency. An automated LLM fine-tuning procedure results in a 30% increase in specification-conformant programs while reducing training time by half.

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

## A    Details of Program-Automaton Conversion

**Executable Program to Abstract Syntax Tree**    Recall that an executable program is a program that consists of a set of predefined keywords and grammar associated with the keywords. Given a program, we first parse it into an AST. We use an existing parsing method with built-in rules for translating programs into ASTs. We present some sample ASTs in table 2.

| AST | *finite state automaton* (FSA) | Note |
|---|---|---|
|  |  | $\sigma = F_C(f_s)$, $\omega = F_C(f_e)$, $f_s \in S, f_e \in E$. "For loop" can be expressed by "while loop." |
|  |  | $\sigma, \omega = F_C(f_s), F_C(f_e)$, $f_s \in S, f_e \in E$. |
|  |  | $\sigma, \omega_1, \omega_2 = F_C(f_s)$, $F_C(f_{e_1}), F_C(f_{e_2})$. For "if-elif-else," we duplicate the "if" node and replace it with "elif." |
|  |  | Running a set of functions sequentially without keywords. $\omega_i = F_C(f_{ei})$ for $i \in [1, 2, 3]$. We can extend it to any number of leaf nodes. |

Table 2: Rules to convert abstract syntax trees to FSA-based representations.

An AST has a set of *tree nodes* and a set of direct transitions between the tree nodes. Each tree node corresponds to a keyword or a function $f \in S \cup E$. A tree node has at most one incoming transition and a set of outgoing transitions connecting to a set of *children* tree nodes. *Root* is a tree node that does not belong to the children of any node, and *leaf* is a tree node whose children are empty.

**Keyword Processor**    The keyword processor is a function mapping an AST with predefined keywords and specified structures to an FSA. It has a set of built-in rules for mapping an AST to an FSA, and we present some sample rules in table 1. The keyword processor cannot handle AST structures beyond the domain of built-in rules.

**Tree to FSA**    So far, we have the AST for the program and the keyword processor, so we can run algorithm 1 to construct an FSA representing the program. First, the algorithm initializes the states, transitions, and labels of an FSA (lines 2-4). Next, it follows a preorder traversal to go through all the tree nodes in the AST (line 9), and it uses the keyword processor to build sub-automata based on the keywords (lines 7). Then, it merges the sub-automata and returns the merged automaton as the final output (lines 11-15).

# B    Additional Experimental Details

We present the programs generated by the language model for the task "bring my backpack back from the lounge" below.

```python
def bring_backpack()_1:
    start_loc = get_current_location()
    go_to("lounge")
    if is_in_room("backpack"):
        while True:
            if is_in_room("person"):
                response = ask("Could you put my backpack in the basket?")
                if response == "Yes":
                    break
            time.sleep(1)
    go_to(start_loc)
```

```python
def bring_backpack_2():
    start_loc = get_current_location()
    go_to("lounge")
    while True:
        if is_in_room("backpack") and is_in_room("person"):
            response = ask(...)
            if response == "Yes":
                go_to(start_loc)
                return
        if not is_in_room("backpack"):
            go_to(start_loc)
            return
        time.sleep(1)
```

We record the average time for generating a program using each baseline models in table 3. The response (program generation) time for GPT-4o-mini and DeepSeek-R1-Qwen-7B is half of the two large models (GPT-4.1 and GPT-4o).

| Model | GPT-4.1 | GPT-4o | GPT-4o-mini | DeepSeek-R1-Qwen-7B |
|---|---|---|---|---|
| Response Time (seconds) | 11.7 | 12.1 | 7.4 | 5.6 |

Table 3: The average time for each model to generate a complete response (program). Note that we query the online models, hence the recorded time includes internet latency. Given that the internet latency is constant, the total response time of GPT-4o-mini and DeepSeek-R1-Qwen-7B is significantly shorter than that of the former two large models.

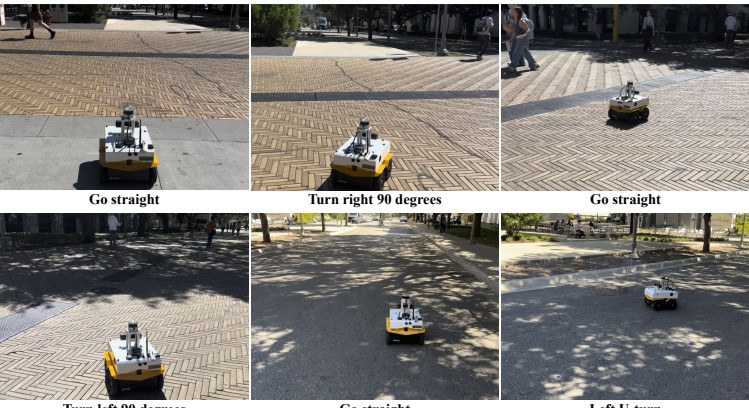

Figure 12: Execution of a composed program that consists of multiple sub-programs. Each sub-program (e.g., go straight, turn left 90 degrees) is formally verified and satisfies the specifications.

## C   Composed Program Execution

Consider we obtain a set of safety-constrained programs for the Jackal outdoor robot by repeating the steps in Section 5.2. The programs include basic driving behaviors such as going straight to approach the stop sign (left of Figure 1), turning left/right (bottom right of Figure 1), U-turn (top right of Figure 1), etc. We compose them into a complex, long-horizon driving task:

> "Always turn right at the stop sign and then go straight,
>
> and make a U-turn if you reach the end of the road."

In Section 4.2, we prove that the composed program from multiple safety-constrained programs also satisfies the safety specifications. We empirically test the composed programs using the outdoor robot. It satisfies the safety specifications during the entire execution.

Theorem 1 allows users or planners to guarantee the composed program satisfies the safety specifications without additional verification to the overall composed program. As the state space of the composed program can be large compared to each individual sub-program, avoiding verifying the overall program significantly reduces the computational complexity.

