# OpenReview forum: "Joint Verification and Refinement of Language Models for Safety-Constrained Planning"
_TMLR — Rejected by TMLR_

### Review · Reviewer_M1J2 · 2025-10-16

**Summary Of Contributions:**

Summary of Contributions

This paper presents a joint verification and refinement framework for improving the safety of large language model (LLM) generated robot programs. The authors propose a method that converts LLM-generated programs into automaton-based representations and verifies them against formal safety specifications using model checking. A key theoretical result, called the compositional verification theorem, proves that any composition of individually verified subprograms will also satisfy safety constraints. This significantly reduces verification complexity for long-horizon tasks.

Building on this, the authors introduce a verification-guided fine-tuning procedure that uses verification outcomes as supervision signals to refine the LLM. Because of the compositional theorem, fine-tuning only needs to cover verified sub-components rather than entire programs, improving both safety compliance and training efficiency.

Empirical evaluations on multiple robot platforms show a 30–50% improvement in the probability of generating safety-compliant programs, about 50% reduction in training time, and good generalization of safety improvements to out-of-domain robots and tasks.

Key Strengths

- Strong theoretical foundation through the compositional verification theorem that provides provable safety guarantees.
- Effective integration of formal verification and LLM fine-tuning to address reliability in safety-critical program generation.
- Demonstrated scalability and transferability across multiple robotic systems.
- Reduced computational and training cost while maintaining or improving safety compliance.

Key Weaknesses

- Experiments are limited to robotic programming, and generalization to other programming domains is not empirically validated.
- Some details of the automaton construction and model checking steps are briefly described, which may limit reproducibility.
- The fine-tuning dataset (100 samples) is small and may not reflect large-scale or real-world diversity.

**Audience:**

Yes

**Audience Explanation:**

The findings would be of clear interest to the TMLR audience, particularly researchers working on safe and verifiable AI systems, LLM program generation, and robotic planning. The paper bridges formal verification and language model refinement, two active research areas within machine learning and AI safety. Its theoretical guarantees and practical demonstrations provide valuable insights for those developing reliable, safety-constrained applications of large language models in real-world systems.

**Claims And Evidence:**

Yes

**Claims Explanation:**

The paper’s claims are supported by clear theoretical and empirical evidence. The compositional verification theorem is mathematically stated and proven, and its implications are validated through experiments on multiple robot platforms. The authors provide concrete examples showing how the verification process identifies unsafe programs that empirical tests would miss. Quantitative results demonstrate consistent improvements in safety compliance and training efficiency, aligning with the claims made in the abstract and introduction. Overall, the evidence presented is accurate, convincing, and clearly connected to the paper’s stated contributions.

**Requested Changes:**

1. Clarify automaton construction process (important for reproducibility).
Provide more detailed explanations or pseudocode for how the abstract syntax tree (AST) is transformed into the finite-state automaton (FSA), particularly how variable naming inconsistencies and complex control structures are handled. This would make the verification pipeline easier to reproduce.

2. Expand empirical validation beyond robotics (recommended).
While the experiments on multiple robots are compelling, the authors could strengthen the paper by testing on additional domains where LLMs generate executable code, such as API-based control or simulated environments. This would demonstrate the generality of the proposed method.

3. Report verification and fine-tuning runtime in more detail (recommended).
Including more explicit runtime comparisons (e.g., how long verification takes per program and per composition) would make the computational benefits more concrete.

4. Improve clarity on dataset creation (minor).
Clarify how the 100 training samples were selected, whether they are unique per task, and if task prompts were generated manually or automatically.

5. Minor writing and formatting edits (optional).
A small language pass to improve flow and consistency, particularly in the methodology section, would enhance readability.

---

> ### Author Response · Authors · 2026-02-14
> **Response to Reviewer M1J2**
>
> We thank the reviewer for the insightful comments and will make every effort to address the concerns.
>
> **Generalization to Other Domains**
>
> This paper is particularly designed for robot planning. We agree that expanding this framework to other domains would be helpful, and we will include this discussion in the future work section.
>
> **Details of the Automaton Construction**
>
> We included the source code in the anonymous GitHub link attached to the abstract. Due to the page limit, we are unable to describe in detail. However, we agree that more details would help reproducibility, and we will include the implementation details in the Appendix.
>
> **Small Fine-tuning Dataset**
>
> We intentionally keep it small to demonstrate that our fine-tuning is data-efficient, i.e., achieve high performance with very limited data.

---

### Review · Reviewer_4J2T · 2026-01-21

**Summary Of Contributions:**

## Summary

This paper introduces a framework for refining language models to
produce formally safe programs, to improve the efficiency of formally
safe execution of LLM-generated plans in the real world.
Overall, the idea is to prompt an LLM to generate programs using a
given API at train time, run formal verification of these programs
against a given safety specification, and fine-tune (via SFT) the LLM
with the verified safe program outputs. Central to the framework is
the authors' argument that it suffices to verify small "sub-programs"
at train time, which can be safely *composed* to create more complex
and verifiably safe behavior at test-time, drastically reducing the
computation complexity of verification.
After fine-tuning, at deployment, the refined LLM is prompted for
programs until they are verifiably safe, which occurs with much higher
likelihood after the proposed fine-tuning approach in robotic experiments.


## Strengths

I like the idea of fine-tuning an LLM to increase the likelihood of
verifiably-safe program outputs&#x2014;I think this could be a very
powerful path forward, and may help increase the safety of AI agents
in the real world.
I also think this fine-tuning procedure largely makes sense and sounds
reasonable and inexpensive.
While no confidence intervals are given, the experimental results
appear to be good, demonstrating that the fine-tuning procedure does
significantly reduce the burden of finding safe programs via test-time
search.


## Weaknesses

Unfortunately, the paper suffers from (what are in my opinion) some
major weaknesses, which I discuss one-by-one below (in no particular order).

**Fine-tuning method**.
Overall, the methodology is very difficult to follow. From what it
boils down to (and correct me if I'm wrong), the method is the
following:

1.  Prompt LLM for program
2.  Use formal verification to decide if program is safe according to
    safety specification
3.  Keep only safe programs, fine-tune on them via standard SFT.

This is a fine-tuning approach that is already quite standard in the
literature (e.g., see STaR, cited below). For any fine-tuning task,
these methods do the following:

1.  Prompt LLM for response
2.  Judge response somehow (e.g., is the solution to the math problem correct)
3.  Keep only good responses, fine-tune on them via standard SFT.

So, your method differs only in the sense that the "judge" is a formal
program verifier. I understand that my job as a reviewer for TMLR is
not to assess novelty, and that's fine; my point here is that this
method could have been explained in a much clearer way, while
simultaneously crediting the closely related work. This is especially
true since the construction of the "judge" (i.e., converting source
code into an FSA and validating the FSA) is itself a standard approach
to formal verification. From the perspective of a ML audience, these
details are not as important, and they make it much more difficult to
identify the actual intervention that you did. Moreover, the notation
and equations used to describe the FSAs and such is very confusing,
and in some cases wrong (see below), which only makes matters worse.

**Composition theorem**.
A central focus in the paper is the idea that it suffices to verify
sub-programs to ensure the safety of programs composed of those
sub-programs&#x2014;this affects the fine-tuning procedure because you then
only need to fine-tune on the generation of much smaller programs.
As I discuss in **Additional Comments**, I am almost certain that this
result is incorrect as stated (Theorem 1), and even if it was corrected, it would
be done at the cost of a *very* strong assumption made on the structure
of the composed programs. Effectively, this assumption would take the
form "if program 1 and program 2 are both safe, then the sequence of
running program 1 followed by program 2 is also safe", at which point,
the proof is trivial (and more importantly, the result is no longer
interesting, because the issue of composition is swallowed by the
assumption). And indeed, if you look at the proof in the
paper, this is basically exactly how it goes.
Similarly, I am also extremely confident that Proposition 2 is
incorrect, and in fact, I believe the incorrectness of Proposition 2
extends even beyond an issue with Theorem 1.

To be clear, I think there are *some* cases of composed programs where
the result holds, but these cases are restricted to ones where the
safety of the composition is obvious *a priori*.

**Experiments**.
As I explain below, the experimental setup in some parts is very
unclear. Most notably, I can't tell if the quantitative results
actually apply to "composed programs"&#x2014;it seems like they don't. In
fact, the paper doesn't really describe how one *should* execute
composed programs in the proposed framework.

**Additional Comments:**

Problem 1 can seemingly be (and almost certainly has been) formulated
in the absence of language models. It's not clear why language models
are even a consideration in Problem 1. Is there something I'm missing?


## Theorem 1

Below are my thoughts about Theorem 1 and its dependents, which
hopefully tracks down what I believe to be the issue and how it could
potentially be "resolved".

I suspect definition 2 isn't complete. Suppose your safety
specification is something like "never press the button while the door
is open". Now, a program $P_1= f_1f_2$ for $f_1$ being "press button"
and $f_2$ being "open door" satisfies the safety constraints. Likewise
$P_2 = f_2$ satisfies the safety constraints. But then you're
seemingly allowed to compose the subroutines of the function
arbitrarily, so that $P_3 = f_2f_1f_2$ is a valid composition of $P_1$
and $P_2$. However, your claim in Problem 2 is that $P_3$ should
satisfy the safety constraints because $P_1$ and $P_2$ do, but $P_3$
directly violates the constraint by opening the door before pressing
the button. So, either the composition rule you defined is too
flexible, the notion of safety constraint is too flexible, or Problem
2 is impossible.

I'm fairly confident that the second assumption of Theorem 1 is not
strong enough. Indeed, looking at the example I gave earlier, I don't
think this second assumption rules out that case. Particularly, I
believe you also need to ensure that $L_2(q')\psi_2$ doesn't contain a
bad prefix.

Regardless of whether or not my concern about the second assumption of
Proposition 1 is true, the proposition itself is quite weak: this is
because, regardless of what is the correct way to state the second
assumption, that assumption is doing literally all of the heavy
lifting in the proof. You design the assumption specifically so that
you can chain safe programs together. *A priori*, it is not at all clear
to me how a system designer can know if this assumption will be
satisfied, so we are back to square one.


## Proposition 2

I have some concerns about the correctness of proposition 2. Suppose
all of the programs $\mathcal{P}_i$ have traces at most $n$ steps
long, and suppose the set of bad prefixes contains some that are $nm$
steps long.
How can you then possibly guarantee that the safety of each $\mathcal{P}_i$
implies the safety of the chain
$\mathcal{P}_1,\mathcal{P}_2,\dots,\mathcal{P}_m$? I suspect the issue
here is again with the structure of the second assumption (it is not
enough to only look at traces that appear before a transition, you
need to look at traces following a transition as well).

**Audience:**

Yes

**Audience Explanation:**

The problem setup is very relevant, certainly a large part of the TMLR
audience ought to be interested in knowing how to improve the safety
of embodied agents / agentic LLMs.

**Broader Impact Concerns:**

As I discussed throughout the review, the paper strongly overclaims their safety
results (though probably not maliciously). The safety applies only to
a very particular subset of programs, not to general sequences of
individually safe programs. Thus, a practitioner using this method can
be severely misled about the safety guarantees it provides.

**Claims And Evidence:**

No

**Claims Explanation:**

See **Weaknesses**. I strongly believe Theorem 1 and Proposition 2 are
incorrect. I also don't see how the authors validated their framework
for the deployment of composed programs, which was a main motivation for
their work.

**Requested Changes:**

## Critical changes

Overall, as I suggest in weaknesses, the description of converting
programs to FSAs / verification should be drastically shortened to
clarify the method.

Moreover, it sounds to me like this method is equivalent to STaR [1], with
reward function goverend by formal verification, and where you
leverage the composition theorem to synthesize shorter supervised
learning inputs/outputs.
As such, I believe some critical literature review is missing.
Recent work, for example [2], has demonstrated also how *negative*
examples can be used to improve fine-tuning; perhaps this could be
quite useful in this setting as well.

Theorem 1 and Proposition 2, at least as explicitly stated in the
manuscript, are almost certainly wrong (see **Additional Comments** for
discussion about these).

Section 4.3 is **way** too informal and missing lots of information.
This is especially true given that, as I understand it, it is the core
contribution of the paper&#x2014;everything else thus far (e.g., building
FSAs from programs) is not new (and the composition results are either
very restrictive or wrong). I have lots of questions:

1.  What do you prompt the language model with during the fine-tuning stage?
2.  What does "given a set [&#x2026;] of task descriptions" mean, and how is
    this encoded / communicated to the language model?
3.  What does it mean to "query a language model $M$ to generate
    executable programs"? Why does it generate more than one program?
    And doesn't it generate source code as opposed to executable programs?
4.  In step 2, shouldn't there be different specifications
    corresponding to different settings / tasks? So shouldn't it be the
    case that not all programs are verified against the same specifications?
5.  Maybe answered by a response to a previous question, but what are the programs
    $P_i$ output by the model *supposed* to be? How do you elicit this
    from the language model?

For the different tasks, the specification of the associated systems
is again very imprecise. For example, in the case of the Jackal
outdoor robot, the text says

> We formally define the system for this robot as follows:
>
> \begin{align*}
> &S = \{pedestrian\_observed()\},\ E = velocity\_publisher(),\ stop(),\\
> &AP_S = \{pedestrian\}, AP_E = \{publish\ velocity, stop\},\\
> &F_C(pedestrian\_observed()) = pedestrian,\ F_C(stop()) = stop,\ F_C(velocity\_publisher()) = publish\ velocity
> \end{align*}

There is a lot that is not "formal" about this. First, none of these
programs are actually defined anywhere. Likewise, none of the
transition behavior is defined anywhere.
While this may sound like I'm begin overly pedantic, issues due to
this non-formalism arise when we try to inspect the language model
responses shown on page 9, it's not actually clear to me if the
programs are correct.
For example, we don't know what
`pedestrian_observed()` is doing: what direction does it look in? Is it
assumed that the attached sensors can detect pedestrians in any angle
that you can send to `velocity_publisher()`?

In Figure 9, clarify how you estimated the
probabilities. Do you use the same prompt(s) across safety
specifications? For each safety specification, do you hold the prompt
fixed? In any case, it would be interesting to see confidence
intervals in this plot. (The same comments apply to Figure 10.)

I don't understand at all how you actually would compose the verified
sub-programs in practice (even under the conditions that the
composition theorem is true). Here are the different ways I can think
of doing this:

1.  Prompt the LLM for the for one program, try to identify
    sub-programs, and verify the subprograms. This sounds hard because
    I don't know how you'd find the subprograms.
2.  Generate a list of verified subprograms during fine-tuning, keep it
    around, and then prompt the LLM for a program defined in terms of
    these sub-programs. This also sounds tough because then you'd have
    to assume some notion of coverage of the sub-programs you see
    during fine-tuning. It's also unclear to me how well the LLM should
    perform in planning over these sub-programs.
3.  Prompt the LLM for one program, and try verifying the whole
    program, ignoring already-verified sub-programs. This seems to
    defeat the purpose, since it could be very expensive to verify the
    whole program at inference time.

I also don't see any results about the performance on these composed
programs, aside from a paragraph in the appendix that doesn't include
any quantitative discussion (nor does it explain how the composed
programs were constructed).


## Minor changes

Throughout the text, citations are often not formatted properly, which
makes reading a little uncomfortable at times. Particularly, `\citet`
should be used for citations that are meant to be part of the
sentence, while `\citep` should be used otherwise.

The notations $AP_S$ and $AP_E$ are a little confusing &#x2013; the $AP$
should not be italicized (it looks like the product of two things).
I suggest `\mathrm{AP}`, or something else that makes it clear that the
$AP$ is one symbol as opposed to the combination of two.

The description of the system is not clear to me. Maybe a diagram
showing the flow of information would help.
For example:

1.  I don't understand the role of the subscribing functions. You say
    it maps text to $\{0,1\}$, what does this do semantically? What
    does this have to do with "API calls"?
2.  The definition of execution functions is even less precise; you say
    an execution function "publish[es] actions for the system to
    execute", but the notion of an "action" hasn't been defined.
3.  You say e.g. "$AP_S$ is a set of atomic propositions corresponding
    to $S$"; what is an atomic proposition, and what does it mean for
    an atomic proposition to "correspond to" the set of subscribing functions?
4.  What is a "safety specification"? It is sort of described below,
    but it is still a little vague.

The statement of Problem 2 is strange; is this a hypothesis/conjecture? A
theorem? It's unclear&#x2014;it is a statement that is either true or
false. A statement left to be proved is not a problem, it is a
conjecture. Based on my previous point, is the idea here that the
problem is to work out the conditions on the composition rule or the
set of safety constraints for which the statement is true?

There are some issues with Definition 3:

1.  I'm pretty sure $T_s$ is not defined correctly, since $\{(q_i, q_j)
       \lvert q_i, q_j\in Q_s\}$ is just $Q_s\times Q_s$.
    I think what you probably want is $T_s\subset Q_s\times Q_s$, where
    $(q_i, q_j)\in T_s$ means that you can transition from $q_i$ to $q_j$.
2.  The "label function" has signature $L_s:Q_s\to 2^{AP}$. What is
    $AP$? I see this is defined after the definition 3, but this is
    uncomfortable for the reader.

In definition 4, my comment about $T_s$ applies to $T_a$.

The "safety property" concept deserves a little more attention; it's
not particularly intuitive and I'm not sure I understand its
motivation (since the word "safe" does not even appear in the
definition&#x2026;). Is my understanding correct that a set of traces is a
safety property only if there are particular finite-length prefixes
that are omitted from the set altogether? Or alternatively, a "danger
property" is a subset of the space of finite length traces, and a
safety property is any set of traces that don't contain prefixes in a
danger property?

In proposition 1, it says "Let $\phi$ describe safety property $P_{\rm
safe}$." I don't understand; what is $\phi$? How does it describe a
safety property? Why not just say "Let $P_{\rm safe}$ be a safety property"?

Remark 1 doesn't read like a remark, it reads like a claim /
proposition. Is the proof of this claim the paragraph above it? This
should be made more precise.

The section title "Demonstration" is confusing for section 5. I was
expecting a demonstration of the presented method. Rather, it seems
that section 5 is demonstrating that existing pre-trained LLMs do
generate unsafe programs that need to be corrected.


## References

1.  Zelikman, Wu, Mu, Goodman. *STaR: Bootstrapping Reasoning with
    Reasoning.* NeurIPS, 2022.
2.  Le Roux et al. *Tapered Off-Policy REINFORCE: Stable and Efficient
    Reinforcement Learning for LLMs*. NeurIPS, 2025.

---

> ### Author Response · Authors · 2026-02-14
> **Response to Reviewer 4J2T**
>
> We thank the reviewer for the insightful comments and will make every effort to address the concerns.
>
> **Justification of Theorem 1 and Proposition 2**
>
> We carefully read the reviewer's comments regarding the correctness of Theorem 1 and Proposition 2. We believe that the reviewer misinterpreted or disregarded Assumption 2 in Theorem 1. And this is the reason that the reviewer doubt the correctness of the theorem.
>
> Intuitively, Assumption 2 states that "the transition from a state in P1 to a state in P2 does not lead to a safety violation." Use Figure 1 as an example, the orange dashed lines are transitions between the subprograms, assumption 2 states that those orange lines will NOT be the source of violation.
>
> In the counterexample provided from the reviewer, $P_3 = f_2f_1f_2$ where $f_2f_1$ violates the safety spec. However, the transition from $f_2$ to $f_1$ also violate assumption 2, hence the theorem holds. Similarly, the induction proof for Proposition 2 holds under assumption 1 and 2 in Theorem 1.
>
> **Novelty of the Composition Theorem**
>
> It is correct that the theorem replying on strong assumtions. However, we claim that
>
> (1) The current assumptions, although strong, are the minimum assumptions for the theorem being hold.
>
> (2) Programs A and B are safe implies the composition of AB safe is trivial when we simply combine them linearly (i.e., finish executing A and then execute B). Additionally, all the counterexamples from the reviewer are single-way, finite traces. However, the actual traces for each (sub)program can be **infinite long** and **infinitely many**. The composition of the programs can also be much more complex than a one-way transition. There can be **infinitely many transitions** from A to B between **any states**, which does not have to be a transition to B after the completion of A.
>
> (3) This work focuses on robot planning (robot program synthesis). The assumptions in Theorem 1 is not hard to be achieved in robot planning tasks. So, they are not overly strong and unrealistic.
>
>
> **Justification on Composed Programs**
>
> Recall that a composed plan consists of a sequence of sub-programs. It can include *any number of repetitions or arrangements* of these sub-plans in any order, making it significantly more complex than a *simple* sequential combination. For example, if we have sub-plans *A(), B(), C(), D()*, a composed plan can be
> ```
> while not ...:
>     A()
>     if ...:
>         B()
>     else:
>         C()
> D()
> ```
> We have presented an execution of a composed program in Appendix C and we will add the corresponding composed program to the Appendix during revision.
>
> **Empirical Study of the Composition Theorem**
>
> We present an example in Appendix C. We state the use of the composition theorem in the *last paragraph of Section 4.2 and 4.3*.
>
> The theorem aims to provide a **theoretical support** to the verification procedure, eliminating the need to verifying long-horizon programs, which **reduces the computational cost of verification**. However, the theorem will not benefit to the compliance rate of generated programs, hence we did not include quantitative analysis on the theorem.
>
> To validate our claim, we compare the average computation time of verifying a composed program versus its corresponding set of sub-programs (supported by our theorem). The numbers in the table below are time in seconds, ran on Apple M4 CPU. Each (sub)program consists of 3~5 states.
>
> | Program Number | 2 | 4 | 8 |
> | - | - | - |- |
> | Time (composed) | 0.07 | 0.18 | 0.45 |
> | Time (ours) | 0.09 | 0.14 | 0.27 |
>
> We will conduct a more comprehensive study and include more details in the revision.
>
> **Fine-tuning Method**
>
> We are not aiming to develop a new fine-tuning algorithm; instead, we provide a new metric--verification outcome--that can be used as the "reward signal" during fine-tuning. This fine-tuning is an add-on to the plan verification framework, presenting an application where the verification results can be used to demonstrate what we can do if the foundation model often fails the verification.

---

### Review · Reviewer_zPSS · 2026-02-02

**Summary Of Contributions:**

The manuscript presents an approach to generating verifiable plans using LLMs. This is done through the following contributions:
- A set of assumptions with which plans/code can be composed safely.
- Finetuning the LLM on verified subprograms.
- Evaluation showing that the resulting planning system indeed efficiently generates verifiable plans.

**Audience:**

Yes

**Audience Explanation:**

This paper presents an approach to tackle the problem of generating safe plans, which is widely recognized as an important problem in robotics.

Furthermore, the system relies on decomposition/divide-and-conquer on the program verification side; having examples of tasks and approaches where this is feasible is useful for the community.

**Broader Impact Concerns:**

I do not believe there are ethical implications of the work that require adding a Broader Impact Statement.

**Claims And Evidence:**

No

**Claims Explanation:**

There are parts of the description of the method and empirical evaluation that require clarification (see requested changes below).

For example, a claim that that is not supported by evidence, is that the manuscript presents 'a method to verify the safety of the LLM-generated programs', if only because the scope of the claim is never stated. Indeed, not all Python programs can be represented with an FSA. I do not believe that this limitation is a critical flaw of the work. In fact, FSAs continue to be widely used in the field of robotics.  However, I do believe this limitation should be discussed and the description of the method should reflect this limitation (e.g., Algorithm 1).

Something that makes the empirical evaluation difficult to understand is the lack of a precise description of the entire system (see the last requested change below).

**Requested Changes:**

1. Not all 'computer programs' can be represented as an FSA. Please add some discussion of this fundamental limitation in the manuscript. E.g.,
	1. What is the class of programs that your system can model as an FSA?
	2. How do you enforce that the LLM generates programs within this class?
2. Definition 1 defines a program as 'a computer program describing a set of function sequences'. Given that this is not a standard definition, the lack of details makes the method difficult to understand. Please expand the description of the formalism. E.g.,
	1) How does this definition relate to the standard understanding of a computer program? Is the idea that the 'sequence of functions' is the sequence of functions that the 'computer program' executes? What is the execution model?
	2) Do you require the 'computer programs' to have any particular form? E.g., SSA? (This is related to requested change 1.)
3. What does 'with its input' and output mean? ("$F_C$ ... maps a function (with its input and output)", page 3). This becomes more confusing in page 4 when $F_C$ receives only a function as input. Please clarify what is the form of $F_C$.
4. What does $L_1 \cup L_2$ mean? (Definition 7)
	1. $L_1$ and $L_2$ are functions. Considering them as sets, and performing the union does not guarantee that the resulting set is a function (e.g., an input could end up mapped to two different images.)
5. (Suggestion) Consider adding a natural language motivation/description to Theorem 1 (and its assumptions).
	1. Question: If I understand, assumption 2 requires that, given you were in a safe trace, you continue being in a safe trace when you transition from automata 1 to automata 2. Why is this not symmetric? I.e., why is there no assumption that mirrors assumption 2 but from automata 2 to automata 1? Figure 1 shows an automata that appears to require this assumption for the proof to work. (Although I acknowledge I might be missing something in the proof).
6. When composing safe programs, how do you prove that the assumptions of Theorem 1 and Proposition 2 hold? Proving that these assumptions hold requires reasoning about the joint automaton; thus (at first glance) verifying the assumptions sounds as hard as verifying the safety of the joint automata in the first place. Please clarify this.
1. What are some of the implementation details for the verifying step? E.g., what software/model do you use for this and what is the effort required on the side of the expert?
3. In page 9, is the output from the LLM a 'subprogram'? Please clarify the role of the LLM in the pipeline and precisely state what the system actually does while searching for a policy. E.g., is the LLM used to 'fill-in' a bigger program? Or is the output of the LLM the full plan/policy?
	- (Followup question) Why is the text model is only finetuned on subprogram generation and not composition? How does the model learn to leverage the compositions of subprograms? Why is finetuning necessary in the first place?
	- What are the details of the evaluation in 6.1? (how does the system autonomously search for a policy?)
	- How does the verification of subprograms come into play if the LLM generates the entire program in a single shot?

---

> ### Author Response · Authors · 2026-02-14
> **Response to Reviewer zPSS**
>
> We thank the reviewer for the insightful comments. We will address the reviewer's comments by following the list under *Requested Changes*, from 1) to 8).
>
> **1) ~ 4) Expressing Programs in FSA**
> We emphasize that our framework is not intended for generic computer programs, but specifically targets high-level robotic planning programs composed of predefined sensing and actuation APIs, as formalized in Definition 1. These programs describe reactive behaviors over a finite propositional abstraction of the environment, which naturally admits finite-state representations and is common in robotics (e.g., task planners and behavior trees). The LLM is constrained implicitly through API grounding and verification-based filtering: it is prompted with fixed sets of system APIs, and only programs that can be parsed, converted to FSAs, and formally verified are retained and used for refinement. Programs outside this class are automatically discarded, and the verification-guided training loop further biases the model toward generating programs within the supported abstraction. This design choice is intentional, as our goal is to provide formal guarantees for robot planning behaviors rather than general-purpose software synthesis.
>
> *Form of F_C:* In our framework, F_C maps an instantiated API call, i.e., a function together with its concrete input arguments and observed Boolean output, to an atomic proposition. For example, a call such as `pedestrian_observed()` returning `True` is mapped to the proposition `pedestrian`. In later sections, we omit explicit arguments and outputs in the notation (writing F_C(f) instead of F_C(f(.))) for brevity.
>
> *L1 U L2 (Definition 7):* It denotes the union of labeling functions over disjoint state sets in the joint automaton.
>
> **5) ~ 6) Assumptions of Theorem 1 and Proposition 2**
> We do have a natural language description of the theorem in the last paragraph of **Section 4.2**. Regarding the asymmetric assumption, we present the minimum assumption needed for the theorem. The symmetric condition is inherently included in Proposition 2, e.g., P1 <--> P2 is equivalent to P1-->P2-->P1, which is covered by Prop 2.
>
> When composing safe programs, we prove the assumption holds by converting each sub-program into an automaton and using a model checker to verify each automaton. The remaining assumption concerns transitions between sub-programs; in our setting, such transitions only occur at program boundaries and are mediated by the same atomic propositions used during sub-program verification. Consequently, any violation introduced at a composition boundary would already appear as a counterexample in at least one sub-program. Practically, this means that verifying each sub-program is sufficient to establish the assumptions required by Theorem 1 and Proposition 2, without constructing or model-checking the full joint automaton.
>
> From a complexity perspective, since composing programs leads to a joint automaton whose state space grows combinatorially with the number of sub-programs, verifying multiple small sub-automata is more efficient than verifying their exponentially larger composed program, which is precisely the motivation behind our compositional approach.
>
> **7) Implementation Details of the Verification Steps**
> We included the source code in the anonymous GitHub link attached to the abstract. Due to the page limit, we are unable to describe in detail. However, we agree that more details would help reproducibility, and we will include the implementation details in the Appendix.
>
> **8) The role of LLM**
> The LLM always generates a **complete executable program** in a single shot. But the complete program can call subprograms. For example:
> ```
> def A():
>     if True:
>         B()
>     else:
>         C()
> ```
> where B and C are subprograms.
>
> Given a task description and a set of system APIs, the LLM generates a full candidate program/policy, which is then formally verified; the system does not perform program filling or symbolic composition during generation.
>
> Fine-tuning focuses on generating safe primitive behaviors because Theorem 1 guarantees that arbitrary compositions of verified programs remain safe; composition itself is handled externally by users or planners, while fine-tuning improves the likelihood that individual generated programs satisfy specifications in the first place.
>
> For each task prompt, the system queries the LLM multiple times with different random seeds to generate candidate programs, verifies each via model checking, and measures the fraction that satisfy all specifications; no additional search or planning is performed beyond this sampling-and-verification loop.
>
> Every LLM-generated program is treated as a subprogram candidate: it is independently verified and stored, and these verified programs can later be reused or composed without re-verification, while fine-tuning increases the probability that future single-shot generations already satisfy safety constraints.

---

### Decision · Action_Editor_rN1c · 2026-03-26

**Recommendation:** Reject

**Additional Comments:**

The central claims regarding the safety of program composition and its practical feasibility under the specified conditions have not yet been sufficiently substantiated or clarified; the scope of the guarantee is limited by the rule-based AST→FSA conversion (which does not cover arbitrary program structures), and the empirical validation of the behavior of composite programs is limited in relation to the main motivation.
These points need to be improved.

A tighter definition, an operational and verifiable treatment of the composition assumptions, a stronger quantitative evaluation of the composed programs, and a clearer positioning are required.

I would like to ask the authors to resubmit the manuscript once all issues have been resolved. This would allow the next group of reviewers to look past the presentation issues and lack of precision and focus on evaluating the value of the proposed framework.

**Audience:**

Yes

**Audience Explanation:**

The reviewers unanimously consider the manuscript to be of interest to the TMLR readership.

The topic is timely, the approach of verification-driven refinement is promising, and the method may prove useful.

**Claims And Evidence:**

No

**Claims Explanation:**

The reviewers unanimously consider the claims and evidence to be insufficiently substantiated.

A considerable number of criticisms were raised regarding the manuscript, most of which were not sufficiently addressed in the rebuttal and require revisions to the manuscript. The criticisms range from concerns about formalism to a lack of details regarding the experimental design.

Many details are missing from the experiments, and the experimental setup itself is criticized.

There are objections regarding the inaccuracy or weaknesses of the theoretical results, particularly in connection with the decomposition of formal verification. Although the authors have addressed this, they have confirmed that the underlying assumption enabling the functionality is so strong that the statement itself essentially constitutes a tautology.

**Resubmission Of Major Revision:**

The authors may consider submitting a major revision at a later time.